# Characterization of *Leuconostoc carnosum* and *Latilactobacillus sakei* during Cooked Pork Ham Processing

**DOI:** 10.3390/foods12132475

**Published:** 2023-06-24

**Authors:** Azra Mustedanagic, Anna Schrattenecker, Monika Dzieciol, Alexander Tichy, Sarah Thalguter, Martin Wagner, Beatrix Stessl

**Affiliations:** 1FFoQSI GmbH—Austrian Competence Centre for Feed and Food Quality, Safety and Innovation, 3430 Tulln, Austria; azra.mustedanagic@vetmeduni.ac.at (A.M.); sarah.thalguter@vetmeduni.ac.at (S.T.); martin.wagner@univie.ac.at (M.W.); 2Unit of Food Microbiology, Department for Farm Animals and Veterinary Public Health, Institute of Food Safety, Food Technology and Veterinary Public Health, University of Veterinary Medicine, 1210 Vienna, Austria; anna.schrattenecker@inext.at (A.S.); monika.dzieciol@vetmeduni.ac.at (M.D.); 3Platform for Bioinformatics and Biostatistics, Department of Biomedical Sciences, University of Veterinary Medicine, 1210 Vienna, Austria; alexander.tichy@vetmeduni.ac.at

**Keywords:** lactic acid bacteria (LAB), brine cultures, microbial limit, genotypic diversity, carbohydrate utilization, food spoilage, food waste

## Abstract

Cooked ham is a popular, ready-to-eat product made of pork meat that is susceptible to microbial growth throughout its shelf life. In this study, we aimed to monitor the microbial growth and composition of nine vacuum-packed cooked ham lots using plate counting until the microbial limit of 7.4 log_10_ AMC/LAB CFU/g was exceeded. Eight out of nine lots exceeded the microbial limit after 20 days of storage. Lactic acid bacteria strains, particularly *Leuconostoc carnosum* and *Latilactobacillus sakei*, prevailed in vacuum-packed cooked ham. *Leuconostoc carnosum* 2 (Leuc 2) and *Latilactobacillus sakei* 4 (Sakei 4) were isolated from raw meat and the post-cooking area of the food processing facility. Carbohydrate utilization patterns of *Leuc. carnosum* PFGE types isolated from raw meat and the food processing environment differed from those isolated from cooked ham. These findings demonstrate how raw meat and its processing environment impact the quality and shelf life of cooked ham.

## 1. Introduction

Cooked ham is a refined ready-to-eat (RTE) product and represents 26% of delicatessen products sold in Europe [1]. The quality of the final product depends on the quality of raw pork meat and processing techniques used, including injection of brine, tumbling, and cooking at a core temperature of 66–75 °C [2,3]. Thermal processing is a crucial step in cooked ham production and has an important impact on microbiota selection [1]. To maintain microbiological and sensory stability during shelf life, vacuum packaging and cold storage are frequently utilized to prevent the growth of potential spoilage microorganisms, eliminating the need for preservatives, antioxidants, and stabilizers [4,5,6]. However, the growth of psychrotrophic, strictly, and facultative anaerobic lactic acid bacteria (LAB) during the cold storage of cooked ham can cause sensory changes (off-flavors and odors, discoloration, and gas and slime formation) and eventually lead to spoilage, rendering the product unsafe for consumption [7,8].

Factors that influence the survival of spoilage organisms during cooked ham production include the microbial quality of the raw meat, the duration of processing, and hygiene in the food processing environment (FPE) [9]. Although the cooking step at a core temperature of ≥70 °C reduces the bacterial load close to the detection limit of the used microbiological methods, some thermoduric LAB species, enterococci, and other microbes may survive this process [10]. Furthermore, there is a risk of product recontamination with spoilage-associated bacteria in the post-cooking area during cooling, slicing, and packaging [11,12]. To counteract this, cooked ham is typically vacuum packed and refrigerated to prevent the growth of Gram-negative spoilage bacteria such as *Pseudomonas* and *Enterobacteriaceae* spp. [5,13]. However, the storage conditions of cooked ham, such as packaging, product composition, hygienic conditions during processing, and storage temperature, can allow LAB to grow to up to 8.0 log colony-forming units (CFU)/g within a few weeks after packaging [8,14,15].

Currently, LAB growth in conventionally processed cooked ham can lead to unacceptable levels and limit its shelf life [16,17,18]. LAB refers to a group of heterogeneous microbial species, such as *Leuconostoc carnosum*, *Leuconostoc gelidum*, *Latilactobacillus sakei*, *Lactobacillus curvatus*, *Carnobacterium divergens*, and *Carnobacterium maltaromaticum*. Some of these species can cause spoilage of the product, while others can contribute to product stabilization as bioprotective cultures with reduced spoilage capacities [10].

Several studies have addressed the cross-contamination of cooked ham from raw meat and food production [10]. Dušková et al. [16] focused specifically on the meat origin of LAB by isolating *Leuconostoc (Leuc.) carnosum* and *Leuc. pseudomesenteroides* from pork carcasses and cooked ham slices. It was observed that once LAB species are introduced into the meat production facility, psychrotrophic LAB species thrive at low temperatures and contribute to cross-contamination of cooked ham batches. Previous studies have reported the isolation of *Leuc. mesenteroides*, *Leuc. carnosum*, *Leuc. gelidum*, *Latilactobacillus* (*Lb*.) *sakei* from FPE and demonstrated their presence in cooked ham as a result of cross-contamination events from raw meat in the pre- and post-cooking areas of the FPE [9,11,12,16,19,20]. Furthermore, a slicing machine has been identified as a critical point for the introduction of LAB into cooked ham [9]. Once present in cooked ham, LAB show tolerance to anaerobic and low-temperature conditions, as well as the influence of sugar-enriched additives used during cooked ham production, which exert pressure on LAB growth and proliferation. The purpose of this study was to evaluate the microbial quality of cooked ham during its storage. Within this context, the monitoring of LAB and aerobic mesophilic counts (AMC) was conducted throughout the shelf life of cooked ham until the maximum acceptable concentration of 7.4 log_10_ CFU/g during cold storage (4 °C) was exceeded. This concentration is considered acceptable according to the AgrarMarkt Austria Marketing GmbH (AMA) quality seal label criteria, which serve as a national guideline for food quality [21]. In addition, the use of brine cultures (*Staphylococcus carnosus*, *Kocuria salsicia*, and *Latilactobacillus sakei*) during the raw meat brining step was also investigated with the potential to extend the shelf life of cooked ham. Raw meat and environmental samples were collected at different stages and screened for the presence of LAB. The most frequently isolated LABs, *Leuconostoc (Leuc.) carnosum* and *Latilactobacillus (Lb.) sakei*, of different origins (i.e., raw meat, production environment, and cooked ham) were identified on strain level. Subsequently, the utilization of common carbohydrates and enzymatic activity were assessed for the frequently isolated pulsed field gel electrophoresis (PFGE) types. Our objective was to use this knowledge to guide production facilities in optimizing their process hygiene practices, which can help prevent food waste and economic losses.

## 2. Materials and Methods

### 2.1. Industrial Ham Production and Processing

Investigated cooked ham was produced in an Austrian meat processing facility using fricandeau meat from the pork leg. Raw meats (RM) were obtained from Austrian slaughterhouses, where they were deboned and transported to meat processing facilities under refrigerated conditions (2–4 °C). In the ham production facility, the standard procedure involved storing the raw meat in the cold room within raw meat delivery boxes (RMDB) for up to five days until processing. A detailed overview of the conditions during the cooked ham production process is presented in Appendix A. Briefly, the manufacturing process included a brining step using 7.0 kg of ice, 64.6 kg of drinking water, 10 kg of 0.5% nitrite curing salt (E250), 7.5 kg of sodium nitrite (E250), 2.0 kg of triphosphates (E451), 0.4 kg of sodium ascorbate (E301), and an 8.5 kg compound mixture including 40% maltodextrin, 12% sugar, 8% dextrose, and a 5.1% spice mixture. The injection rate was 12%. After brining, the meat was transferred to tumbling machines and massaged, followed by a rest period to allow the diffusion of the brine into the meat. Subsequently, the tumbled ham was wrapped in a cellulose casing, molded, and transferred to the cooking chamber. During the cooking step, the temperature was gradually increased until the ham reached a core temperature of 72 °C, which was maintained for 30–60 min. After cooking, the ham was placed on a trolley and rapidly cooled down to a core temperature of 2–4 °C. The cellulose casing was removed, then the cooked hams were placed in vacuum-shrink foil bags (Sealed Air Corporation of Cryovac Inc., Aurora, IL, USA) using a Cryovac VS90 automatic belt feed vacuum packing machine (Cryovac Inc., Aurora, IL, USA) and subsequently vacuum packed with a Cryovac Sealed Air, S.R.L. Type ST98 (Cryovac Inc., Aurora, IL, USA) vacuum packaging machine at 4 °C. The cooked ham was stored at 4 °C in a temperature-controlled cold room. The commercial shelf life of vacuum-packed cooked ham is defined by the Austrian meat processing facility, which sets a sell-by date of up to 40 days after the day of packaging.

### 2.2. Study Design and Sampling

The local cooked ham production factory in Austria provided raw meat, cooked ham, and swab samples from the food processing environment (FPE) under real processing conditions between February and September 2021. The samples were categorized into raw meat (RM), FPE, and cooked ham (CH), and a detailed sample overview is provided in Table 1. Analysis using culture-dependent methods was conducted on nine vacuum-packed cooked ham lots (1–9) during cold storage at 4 °C. The study was divided into three phases.

In the first phase, the microbial status of cooked ham during cold storage was evaluated by examining only the end product of lots 1 to 3. Consequently, the investigation of lots 1 to 3 did not encompass the raw meat used in the production of cooked ham.

The second phase involved the microbial analysis of vacuum-packed cooked ham lots 4 to 7, including the raw meat used in their production. The aim was to capture the rapid growth phase of the cooked ham microbiota until it exceeded the microbial limit of 7.4 log_10_ AMC/LAB CFU/g.

Phase three focused on cooked ham lots 8 to 9, where the raw meat was treated with brine cultures (*Staphylococcus carnosus*, *Kocuria salsicia*, and *Latilactobacillus sakei*) (Wiberg, FRUTAROM Savory Solutions Austria GmbH, Salzburg, Austria) during the brining step. Both the cooked ham batches with and without the brine cultures were microbiologically analyzed throughout the shelf life.

The microbiological investigation of the vacuum-packed cooked hams (CH) comprised a total of 90 samples (lots 1 to 9). The samples were analyzed between 0 and 33 days, except for lots 8 and 9. The aim was to determine when the microbial limit of 7.4 log_10_ AMC/LAB CFU/g was exceeded during cooked ham cold storage. Duplicate samples from lots 1 to 3 were assessed at three timepoints: During phase one of the study, the general microbial status of cooked ham was assessed; therefore, the cooked ham samples were investigated at three different timepoints, including days 0, 11, and 20 of cooked ham storage. In phase two of the study, lots 4 to 7 were analyzed at shorter time intervals to capture the fastest growth phase of cooked ham microbiota until the microbial limit was exceeded. Therefore, there was a slight variation in sampling days until the microbial limit was reached. Cooked ham lots 8 to 9 were incubated until storage day 33 to describe the changes in microbial growth dynamics after exceeding the microbial limit. During the second and third phases of the study, swab samples were taken from the FPE to assess the influence of the hygiene conditions in the pre- and post-cooking areas of the FPE on the microbiota of cooked hams. Swab samples were taken by a trained employee of the cooked ham production facility along the processing line during the production of lots 7 to 9. The surfaces (10 × 10 cm^2^) in FPE were sampled using sterile polyurethane sponges (World Bioproducts, Woodinville, WA, USA) and subsequently placed into a sterile plastic bag.

In the pre-cooking area, the swab samples of brine (B, *n* = 5) (200 mL) were collected in sterile containers with screw caps during the production of lots 7 to 9. To gain more detailed insight into meat-derived cross-contamination in the pre-cooking area during the production of lot 9, samples were taken from raw meat delivery boxes (RMDB; *n* = 1) and ham after tumbling (HAT; *n* = 4). In addition, swab samples were taken from the tumbler after sanitation (TAS; *n* = 2) before the start of production of lots 7 and 9 in order to estimate the cleaning efficiency of the tumbler, which was routinely performed once a week. In the post-cooking area, swab samples were taken from the cellulose casing used for the molding and cooking of the ham (PC; *n* = 5), the trolley used to cool down the cooked ham (T; *n* = 5), the cutting board (CB; *n* = 5), and the personnel gloves (G; *n* = 5) during the packaging of the end product during the production of lots 7 to 9.

The raw meat and FPE samples were processed immediately upon arrival. The Austrian meat processing facility supplied vacuum-packed cooked ham upon completion of each production lot. Subsequently, the cooked hams were stored in a refrigerated room maintained at a temperature of 4 °C until the day of analysis, when they were opened.

### 2.3. Microbiological Analysis and pH Measure

The quantification of aerobic mesophilic count (AMC) and LAB counts in raw meat, food processing environments, and cooked ham was carried out according to ISO reference methods (ISO 4833-2:2013, ISO 15214:1998) [22,23]. The counts of *Enterobacteriaceae* (EB) and *Pseudomonadaceae* (PS) in raw meat were determined according to ISO 21528-2:2017 [24]. Raw meat, ham after tumbling, and cooked ham samples (25 g each) were diluted in duplicate in 225 mL of buffered peptone water (BPW) (Biokar Solabia Diagnostics, Pantin, France) and homogenized in a laboratory mixer (Stomacher^®^ bag; Seward Ltd., Worthing, West Sussex, UK) for 180 s. Environmental sponge samples (World Bioproducts, Woodinville, WA, USA) were diluted in 50 mL of BPW and manually homogenized for 1 min. Brine (200 mL) was centrifuged at 8000 rpm for 30 min at 4 °C (Thermo Scientific, Sorvall Lynx 4000 centrifuge, Thermo Fisher Scientific Inc., Waltham, MA, USA) and the pellet was diluted in 45 mL BPW. Brine cultures (0.1 g) were diluted in 10 mL of BPW and vortexed until dissolved. Subsequently, serial ten-fold dilutions were prepared up to dilution −10^10^ in BPW. The dilutions (100 µL) were plated on Trypto-Caseine Soy Agar supplemented with yeast extract (TSAY) plates (Biokar Solabia Diagnostics, Pantin, France) and All-Purpose Tween (APT) agar plates (Merck, Darmstadt, Germany). For the first dilution, 1 mL of the sample was plated on TSAY, APT, and VRBG. The AMC count was determined on TSAY agar plates that were incubated aerobically at 30 °C for 48 h. The LAB counts were determined on APT agar plates that were incubated microaerobically (Thermo Scientific CampyGen™ 2.5 L Sachet, Oxoid Ltd., Hampshire, UK) at 30 °C for 48 h. The *Enterobacteriaceae* (EB) and *Pseudomonadaceae* (PS) counts were determined on Violet Red Bile Glucose (VRBG) agar plates (Merck KgaA, Darmstadt, Germany) after aerobic incubation at 30 °C for 48 h. The EB and PS colonies on VRGB agar were differentiated by their ability to ferment glucose, resulting in pink colonies with or without precipitation and pale colonies for PS. Presumptive EB and PS isolates were confirmed using an oxidase reaction (BioMerieux, Marcy I’Etoile, France) and subjected to 16S rRNA gene sequencing (as described in Section 2.4). The minimum and maximum limits for the determination of the AMC, LAB, EB, and PS in the samples ranged between 10 and 300 CFU. During the analysis of cooked ham lots 8 to 9, the pH value was determined by taking at least three measurements using a pH meter (Professional Portable pH Meter, Hanna Instruments Inc., Woonsocket, RI, USA).

### 2.4. Isolate Collection and Identification

Up to five representative colonies from each TSAY (*n* = 270), APT (*n* = 280) or VRBG (*n* = 58) agar were subcultured on the corresponding medium. The purified isolates (*n* = *156* from raw meat, *n* = 223 from FPE; *n* = 229 from cooked ham) were cryopreserved at −80 °C in Brain Heart Infusion Broth (Biokar Solabia Diagnostics) supplemented with 25% (v/v) glycerol (Merck KgaA). The isolate list is provided in Appendix A. DNA extraction was performed according to a protocol published by Walsh et al. [25]. Briefly, bacterial material (10 µL) from the agar plate was resuspended in 100 µL of 0.1 M Tris-HCl pH 7 buffer (Sigma Aldrich, St. Louis, MO, USA) and centrifuged at 15,000× *g* for 5 s (Eppendorf Centrifuge 5425, Hamburg, Germany). Subsequently, 400 µL of Chelex 100-Resin (BioRad, Hercules, CA, United States) was added to the bacterial suspension and heated at 100 °C for 10 min on the block heater (Thermo ScientificTM block heater, Thermo Fischer Scientific Inc., Waltham, MA, USA). The suspension was subsequently centrifuged at 15,000× *g* for 5 s (Eppendorf Centrifuge 5425, Hamburg, Germany), and the supernatant (100 µL) was transferred to maximum recovery tubes (Corning Incorporated Life Sciences, Reynosa, Mexico) and stored at −20 °C until analysis. Identification of bacterial isolates was carried out by partial sequencing of the 16S rRNA gene using universal primer pairs 616F (5′-AGAGTTTGATYMTGGCTC-3′) and 1492R (5′-GGYTACCTTGTTACGACTT-3′) (both Microsynth AG, Blagach, Switzerland) as previously described [26,27]. A single PCR reaction (45 µL) contained diethylpyrocarbonate (DEPC)-treated water (Sigma Aldrich, St. Louis, MO, USA), 1 × buffer, 2 mM MgCl_2_, 200 nM forward and reverse primers, 20 mM dNTP mix, 2 U of Platinum Taq DNA polymerase (Platinum™ Taq DNA Polymerase, DNA-free, Thermo Fisher Scientific Inc., Waltham, MA, USA), and 5 µL genomic DNA. The DNA amplification was performed in a T100TM Thermal Cycler (Bio-Rad, Hercules, CA, USA). Thermocycling conditions were 95 °C for 5 min, 35 cycles at 94 °C for 30 s, 52 °C for 30 s, 72 °C for 60 s, and final elongation at 72 °C for 7 min. When 16S rRNA gene PCR yielded negative results, the presumptive fungi colonies were microscopically examined and then submitted to internal transcribed spacer 2 (ITS2) region sequencing. Presumptive fungi isolates were confirmed by sequencing the internal transcribed spacer 2 (ITS2) using primers ITS3 (5′-GCATCGATGAAGAACGCAGC-3′) and ITS4 (5′-TCCTCCGCTTATTGATATGC-3′) [28]. A single PCR reaction (25 µL) consisted of diethylpyrocarbonate (DEPC) treated water (Sigma Aldrich), 1 × buffer, 2 mM MgCl2, 200 nM forward and reverse primers, 20 mM dNTP mix, 0.63 U of Platinum Taq DNA polymerase (Platinum™ Taq DNA Polymerase, DNA-free, Thermo Fisher Scientific Inc., Waltham, MA, USA), and 1 µL template genomic DNA. The DNA amplification was performed in a T100TM Thermal Cycler (Bio-Rad, Hercules, CA, USA) at 95 °C for 5 min, 30 cycles at 94 °C for 40 s, 56 °C for 40 s, 72 °C for 60 s, and final elongation at 72 °C for 7 min. PCR products were previously evaluated by 1.5% agarose gel electrophoresis containing 1 × Tris-Acetate-EDTA buffer (TAE) and 3.5 μL peqGREEN DNA gel stain (VWR International, Radnor, PA, USA), at 120 V for 30 min. For the fragment length comparison, the DNA standard Thermo Scientific™ GeneRuler™ 1 kbp (Thermo Fisher Scientific Inc., Waltham, MA, USA) was applied. Subsequently, the obtained bacterial and fungal PCR amplicons were sent for purification and Sanger sequencing (LGC Genomics GmbH, Berlin, Germany). The bacterial genomic DNA PCR amplicons were sequenced using the 1492R (5′-GGYTACCTTGTTACGACTT-3′) primer from LGC Genomics (LGC Genomics GmbH, Berlin, Germany). The fungi PCR fragments were sequenced using the ITS4 (5′-TCCTCCGCTTATTGATATGC-3′) primer. The quality evaluation of the nucleotide sequences was performed using Finch TV 1.4.0 (Geospiza Inc. Seattle, WA, USA; https://digitalworldbiology.com/FinchTV, accessed on 25 March 2021). For bacterial datasets, the Nucleotide BLAST (Basic Local Alignment Search Tool) algorithm from the National Centre for Biotechnology Information was used for taxonomy assignment (https://blast.ncbi.nlm.nih.gov/Blast.cgi, accessed on 30 March 2021). For fungi datasets, UNITE was used for taxonomy assignment (https://unite.ut.ee/, accessed on 30 July 2021). Sequences were assigned to genus or species level according to best matches and highest similarities (similarity cut-off ≥98%). The partial 16S rRNA gene sequence data of the isolates were deposited in the GenBank database under accession numbers OP263127–OP263616, while the fungi sequence data were under accession numbers OQ940410–OQ940442 (https://www.ncbi.nlm.nih.gov/genbank/, accessed on 9 May 2023).

### 2.5. Molecular Subtyping of Leuconostoc carnosum and Latilactobacillus sakei

The *Leuconostoc (Leuc.) carnosum* (*n* = 144) and *Latilatcobacillus (Lb.). sakei* (*n* = 88) isolates were identified on a strain level using a molecular subtyping method. The detailed list of *Leuc. carnosum* (raw meat, *n* = 2; FPE, *n* = 20; cooked ham, *n* = 122) and *Lb. sakei* (raw meat, *n* = 7; FPE, *n* = 26; cooked ham, *n* = 55) selected for molecular subtyping is provided in Appendix A. The *Leuc. Carnosum* and *Lb. sakei* isolates intended for pulse field gel electrophoresis (PFGE) analysis were grown on APT agar at 30 °C for 48 h under microaerobic conditions (Thermo Scientific CampyGenTM 2.5L Sachet, Thermo Fisher Scientific Inc., Waltham, MA, USA). Subsequently, bacteria were subcultured by inoculating 5 mL of de Man, Rogosa and Sharp (MRS) broth (Oxoid Ltd., Hampshire, UK) and grown overnight at 30 °C under microaerobic conditions. The preparation of plugs and the DNA restriction digestion were carried out as previously described with modifications [19,29]. Briefly, for plug preparation, cells were harvested from 2 mL of MRS broth after they reached an OD_600_ 0.9–1.5 (Oxoid Ltd., Hampshire, UK) by centrifugation at 8000× *g* for 5 min. The resulting pellet was resuspended in 1 mL of ice-cold PIV buffer (0.01 M Tris-HCl pH 7, 1 M NaCl; Sigma-Aldrich Corp., St. Louis, MO, USA). For the cell lysis, 240 µL of the bacterial suspension was transferred into a separate tube and mixed with 60 µL of lysozyme (10 mg/mL; Sigma-Aldrich Corp, St. Louis, MO, USA). The suspension was incubated for 30 min at 56 °C. After cell lysis, 25 µL of proteinase K (20 mg/mL) (Roche Diagnostics GmbH, Mannheim, Germany) was added to each suspension. Subsequently, the bacterial suspension (325 µL) was mixed with 1.2% (w/v) SeaKem Gold agarose (Lonza Group, Basel, Switzerland), which had previously been prepared in PIV buffer, and dispensed into the molds and left to solidify at room temperature. The second cell lysis step was performed by transferring the solidified plugs to 5 mL of cell lysis buffer (50 mM Tris, 50 mM EDTA, pH 8, 1% lauroylsarcosine, 0.1 mg/mL proteinase K; Sigma-Aldrich Corp., St. Louis, MO, USA) that were incubated overnight in shaking water bath at 54 °C (120 rpm). After cell lysis, the plugs were washed twice in 10 mL ddH2O and three times in 10 mL Tris EDTA buffer (10 mM Tris, 1 mM EDTA, pH 8.0; Sigma Aldrich Corp.) for 10 min at 54 °C each. The restriction of *Leuc. carnosum* genomic DNA was performed with *ApaI* (0.25 U/µL; Thermo Fischer Scientific Inc., Waltham, MA, USA) at 25 °C for 4.5 h. The genomic DNA of *Lb. sakei* was digested with *AscI* (0.25 U/µL; Thermo Fischer Scientific Inc.) at 37 °C for 4.5 h. The plugs were loaded onto a 1% SeaKem Gold Agarose gel, and PFGE was performed in a CHEF DR III system (Bio-Rad Laboratories Inc., Hercules, CA, USA) in 0.5 × Tris borate EDTA (TBE) running buffer (45 mM Tris, 45 mM borate, 1 mM EDTA; Sigma-Aldrich Corp., St. Louis, MO, USA) for 22.5 h at 6 V/cm with a linear ramping factor and pulse times from 4.0 to 40.0 s at 14 °C and an included angle of 120°. After gel electrophoresis, the gel was stained with ethidium bromide (Sigma-Aldrich Corp.) and digitally photographed with a Gel Doc 2000 (Bio-Rad Laboratories, Inc., Hercules, CA, United States). The TIFF images were normalized with the BioNumerics 6.6 software package (Applied Math NV, Sint-Martens-Latem, Belgium) using the universal *Salmonella* ser. Braenderup H9812 standard. Pattern clustering utilized the unweighted pair group method with arithmetic mean (UPGMA) and the dice correlation coefficient with a position tolerance of 1.5%. PFGE types were considered identical when the patterns were indistinguishable. The Simpson’s index of diversity was calculated using the Comparing Partitions online tool (http://www.comparingpartitions.info/index.php?link=Home, accessed on 6 June 2022).

### 2.6. Carbohydrate Utilization and Enzymatic Activity of Isolates

Isolates from raw meat (*Leuc. carnosum*, *n* = 2; *Lb. sakei*, *n* = 5), FPE (*Leuc. carnosum*, *n* = 4; *Lb. sakei*, *n* = 19), and cooked ham (*Leuc. carnosum*, *n* = 45; *Lb. sakei*, *n* = 24) as shown in the isolate list (Appendix A) with identified PFGE types were tested for the utilization of 49 carbohydrates using API 50 CHL (API System, BioMerieux, Marcy I’Etoile, France), according to the manufacturer’s instructions. *Leuc. carnosum* and *Lb. sakei* isolates recovered at the beginning (<6 days), during (9 to 11 days), and at the end (>20 days) of cooked ham storage were analyzed using API 50 CHL. The LABs were grown as described in Section 2.4. The isolates were inoculated into API strips and incubated for 48 h at 30 °C. Subsequently, the acid production from the supplied carbohydrates was determined as described by the manufacturer. For the evaluation of carbohydrate utilization patterns, we categorized the PFGE types into different biochemical profiles (BP) in dependence on their specific carbohydrate fermentation patterns and calculated the percentage of carbohydrates utilized per PFGE type.

The enzymatic activity of the identified distinct *Leuc. carnosum (n* = 14 isolates) *and Lb. sakei (n* = 12 isolates) PFGE types was analyzed using API ZYM (Bio Merieux, Lyon, France) as described by the manufacturer. The isolates selected for the API ZYM test are shown in the isolate list in Appendix A. The LABs were grown as described in Section 2.4. Subsequently, the suspension was spotted (45 µL) into wells and incubated at 25 °C for 4 h. Then, one drop of each of the kit reagents, ZYM-A and ZYM-B, was added to each well. The wells were incubated for 5 min, allowing the reactions to develop. The enzymatic activity was determined as described by the manufacturer.

### 2.7. Statistics

Using the program IBM SPSS v28, a binary logistic regression analysis was applied to estimate the probability of reaching the limit of 7.4 log_10_ CFU/g (AMC and LAB) using storage day as the predictor. A *p*-value below 5% (*p* < 0.05) was seen as significant.

## 3. Results

### 3.1. Microbial Characterization of Raw Meat and Cooked Ham during Storage

To assess the level of microbial contamination in raw meat utilized for cooked ham production, the counts of AMC, *Enterobacteriaceae* (EB), and *Pseudomonadaceae* (PS) were evaluated in lots 4 to 9. The AMC load in raw meat ranged from 3.4 ± 0.05 to 5.9 ± 0.43 log_10_ AMC CFU/g, with the highest counts detected in lots 4, 7, and 9 (Table 2). The counts of LAB ranged from 2.6 ± 0.28 to 5.8 ± 0.07 log_10_ CFU/g, with the highest counts detected in lot 9. The EB counts in raw meat from lots 4, 5, 7, and 9 ranged from 1.1 ± 1.29 to 4.7 ± 0.13 log_10_ CFU/g, while the PS counts from lots 5, 6, 7, and 9 ranged from 1.7 ± 0.40 to 4.7 ± 0.33 log_10_ CFU/g. No EB growth was detected in raw meat lots 6 and 8, while no PS growth was detected in lots 4 and 8. Of particular interest was the isolation of *Leuc. carnosum* and *Lb. sakei* from raw meat, as these were two primary bacterial species identified during the cooked ham storage. *Leuc. carnosum* was isolated from raw meat lots 5 and 7 (16.7%; *n* = 2/12; Table 2), while *Lb. sakei* was isolated in lots 4 and 8 (33.3%; *n* = 4/12; Table 2). Additionally, other frequently isolated genera from raw meat were spoilage bacteria belonging to the genera *Aeromonas*, *Brochothrix*, *Carnobacterium*, *Pseudomonas*, *Staphylococcus*, and *Streptococcus* (Appendix A). Yeasts, including *Candida*, were isolated from raw meat in every lot except for lot 5, whereas *Yarrowia* was isolated only from lot 6 (Appendix A).

A total of 90 samples from nine different cooked ham lots (1 to 9) were microbiologically evaluated from the date of packaging (day 0) until the point when the maximum acceptable microbial limit of 7.4 AMC/LAB log_10_ CFU/g was exceeded during the cold storage at 4 °C (Table 3). At the beginning of the storage (day 0), the microbial counts were below the detection limit (<1.0 log_10_ CFU/g) for most lots except for lot 1 (3.5 ± 0.01 AMC and LAB log_10_ CFU/g) and lot 4 (1.3 ± 0.43 AMC log_10_ CFU/g) (Table 3). During the storage period, eight of nine lots exceeded the microbial limit on day 20, with lot 6 being the only exception, which remained below the detection limit until day 20 of storage (1.7 ± 0.57 AMC log_10_ CFU/g). Specifically, lots 1, 4, and 5 exceeded the limit on day 11 of storage, while lots 7, 8, and 9 exceeded the limit on day 15 of storage. Lots 2 and 3 exceeded the limit on day 20 of storage. Microbial counts for AMC and LAB reached a plateau between days 15 and 33 of storage, ranging between 7.9 and 9.2 log_10_ CFU/g for AMC and 7.8 and 9.1 log_10_ CFU/g for LAB.

The probability of achieving 7.4 log_10_ CFU/g AMC reaches 50% on day 11 and 100% on day 15. However, logistic regression showed no significant OR. When the same is performed for LAB, the probability of achieving 7.4 log_10_ CFU is 19% on day 11, 58% on day 15, and 93% on day 20, with a significant OR of 1.57 (*p* = 0.002). During the storage of cooked ham lots 1 to 9, the predominant microbial flora consisted of 62.2% *Leuc. carnosum* (*n* = 56/90) and 37.8% *Lb. sakei* (*n* = 34/90). Other species of LAB (7.8%; *n* = 7/90), including *Latilactobacillus graminis*, *Leuconostoc mesenteroides*, and *Weissella viridescens*, were only isolated at the beginning of the storage (Appendix A). In addition, other non-LAB bacteria (13.3%; *n* = 12/90) belonging to the genera *Pseudomonas*, *Kocuria*, *Corynebacterium*, *Bacillus*, and *Staphylococcus* were also isolated at the beginning of the storage (Appendix A). Yeasts, including *Cutaneotrichosporon* and *Filobasidium*, were isolated at the beginning of storage lot 9 (Appendix A).

### 3.2. Effect of Brine Cultures on Cooked Ham Shelf Life

The microbiological evaluation of the potential of brine cultures to prolong shelf life was conducted during the production of lots 8 and 9 (*n* = 64 samples). In the brining step, raw meat (*n* = 32) was treated with brine cultures, including *Staphylococcus carnosus*, *Kocuria salsicia*, and *Latilactobacillus sakei*, and compared to the batch (*n* = 32) without brine cultures. The results showed that in the case of lot 8, the batch with brine cultures exceeded the microbial limit on day 20 of storage, while the batch without brine cultures exceeded the limit on day 15 (Figure 1a). In addition, the pH value during the storage of cooked ham lots 8 and 9 was also monitored (Figure 1b). For lot 8 with brine cultures, the initial pH value was 6.61 ± 0.24 and decreased to a pH value of 5.88 ± 0.08 on day 33 of storage. For lot 8 without brine cultures, the initial pH value was 6.06 ± 0.08, and it decreased to a pH value of 5.62 ± 0.23 on day 33 of storage.

In lot 9, the batch with brine cultures exceeded the microbial limit on day 11 of storage, while the batch without brine cultures exceeded the limit on day 15 (Figure 1a). For lot 9 with brine cultures, the initial pH of 6.05 ± 0.04 increased to a pH value of 6.51 ± 0.02 during the cooked ham storage (day 11), and it decreased to a pH value of 5.98 ± 0.05 on day 33 of storage (Figure 1b). For lot 9 without brine cultures, the initial pH value of 5.91 ± 0.12 increased to a pH value of 6.14 ± 0.01 during cooked ham storage (day 11) and decreased to a pH value of 5.54 ± 0.31 on day 33 of storage.

During the storage of cooked ham lots with brine cultures, *Leuc. carnosum* was isolated from 50.0% (*n* = 16/32) of the samples, while *Lb. sakei* was isolated from a single sample on day 33 of storage in lot 8 (3.1%, *n* = 1/32) (Appendix A). In lot 9 with brine cultures, other LAB species (9.4%, *n* = 3/32) and non-LAB bacteria (6.3%, *n* = 2/32) were sporadically identified during the storage, including *Carnobacterium* and *Leuconostoc* genera, as well as *Microbacterium*, *Pseudomonas*, and *Staphylococcus* genera (Appendix A).

### 3.3. Microbial Load and LAB Occurrence in the Food Processing Environment

The microbial examination of samples from the food processing environment (FPE) was conducted to identify the presence of LAB in both the pre- and post-cooking areas (Table 4). The microbial loads on surfaces in the pre-cooking area ranged from 1.0 to 3.1 log_10_ CFU/cm^2^ for AMC and LAB. The brine had microbial loads ranged from 1.2 to 3.8 log_10_ AMC CFU/mL and 2.7 to 3.9 log_10_ LAB CFU/mL. The brine cultures used during the brining step used for production of lots 8 and 9 had a concentration of 10.5 to 13.1 log_10_ AMC CFU/mL and 10.8 to 13.1 log_10_ LAB CFU/mL. The microbial counts in ham after tumbling ranged from 4.6 to 6.6 log_10_ AMC CFU/g and 3.9 to 6.5 log_10_ LAB CFU/g. In the post-cooking area, the microbial counts were low (1.0 to 2.4 log_10_ AMC CFU/cm^2^ and 1.5 to 2.3 log_10_ LAB CFU/cm^2^), with the highest counts identified on the cutting board.

Regarding the LAB in the FPE, both *Leuc. carnosum* and *Lb. sakei* (each 35.3%; *n* = 12/34) were isolated in the pre- and post-cooking areas, including the tumbler after sanitation, the ham after tumbling, the cellulose casing, the cutting board, and the gloves of the personnel (Table 4). In addition, *Leuc. carnosum* was found on the trolley in the post-cooking area, while *Lb. sakei* was identified as one of the species contained in the brine. During the cooked ham production, increased microbial diversity was observed on the cellulose casing, brine, and cutting board. Spoilage bacteria commonly associated with raw meat (e.g., *Brochothrix*, *Carnobacterium*, *Pseudomonas*, and *Staphylococcus* spp.) were isolated in the pre-cooking area, while mainly *Kocuria*, *Micrococcus*, *Pseudomonas*, and *Psychrobacter* spp. were isolated in the post-cooking area (Appendix A). Yeast *Candida* was identified in the post-cooking area, including the cellulose casing, cutting board, and personnel gloves during the packaging of the end product (Appendix A).

### 3.4. Leuc. carnosum and Lb. sakei Strain-Level Characterization

Molecular strain-level analysis of the commonly isolated *Leuc. carnosum* and *Lb. sakei* LABs at different stages of cooked ham production (RM, FPE, and cooked ham) revealed that particular pulsed field gel electrophoresis types (PFGE types) were responsible for contaminating the final product. For the 144 *Leuc. carnosum* isolates (RM, *n* = 2; FPE, *n* = 20; cooked ham, *n* = 122), PFGE *ApaI* profiling resulted in 12 PFGE types and two subtypes, with a Simpson’s diversity index of 0.668 (CI 95%, 0.494–0.782) (Figure 2a). In the pre-cooking area, diverse *Leuconostoc carnosum* (Leuc) PFGE types (Leuc 2, Leuc 6, Leuc 7, Leuc 8, Leuc 11, and Leuc 12) were present. In the post-cooking area, four different PFGE types (Leuc 1, Leuc 2, Leuc 3, and Leuc 8) were identified. The two most frequently isolated PFGE types in the post-cooking area, Leuc 2 and Leuc 3, were also the most abundant PFGE types in cooked ham. The PFGE type Leuc 2 (50%, *n* = 72/144) was identified in raw meat, the post-cooking area (cellulose casing and cutting board), and all cooked ham lots, while the PFGE type Leuc 3 (27.8%, *n* = 40/144) was isolated from all tested surfaces in the post-cooking area and six cooked ham lots (1, 3, 4, 7, 8, and 9). Additionally, five other PFGE types (Leuc 1, Leuc 4, Leuc 5, Leuc 9, and Leuc 10) and three subtypes (Leuc 1 ST, Leuc 5 ST, and Leuc 9 ST) were sporadically isolated during the cold storage of cooked ham lots (1, 2, 4, 8, and 9).

For the 88 *Lb. sakei* isolates (RM, *n* = 7; FPE, *n* = 17; brine cultures, *n* = 9; cooked ham, *n* = 55), subtyping resulted in 11 unique *AscI* profiles and one subtype, with a Simpson’s diversity index of 0.839 (CI 95%, 0.809–0.870) (Figure 2b). In the pre-cooking area, eight *Latilactobacillus sakei* (Sakei) PFGE types (Sakei 2, Sakei 3, Sakei 4, Sakei 5, Sakei 7, Sakei 9, Sakei 10, and Sakei 11) and one subtype (Sakei 4 ST) were identified. In the post-cooking area, four different PFGE types (Sakei 3, Sakei 4, Sakei 8, and Sakei 10) were identified. The PFGE type Sakei 4 was the most frequently identified PFGE type (25.0%, *n* = 22/88), isolated from raw meat, the post-cooking area (cellulose casing and cutting board), and six cooked ham lots (2, 4, 5, 7, 8, and 9). The PFGE type Sakei 3 was the second most frequent type (14.8%; *n* = 13/88), isolated from raw meat, the cutting board, and five cooked ham lots (1, 2, 4, 5, and 8). The PFGE type Sakei 7 was identified as one of the isolates present in brine cultures. The PFGE type Sakei 7 (20.5%; *n* = 18/88) indicated dissemination along the cooked ham processing line (raw meat, brine, and ham after tumbling) during the production of lot 8 and 9 batches with brine cultures. Other PFGE types (Sakei 1, Sakei 5, Sakei 6, and Sakei 10) were sporadically identified during the storage of cooked ham lots (1 and 7).

### 3.5. Biochemical Characterization of Leuc. carnosum and Lb. sakei Isolates

In order to provide an in-depth characterization of LAB isolates, we conducted an analysis to determine whether the *Leuc. carnosum* and *Lb. sakei* isolates could ferment various types of sugar, such as mono-, di-, and trisaccharides, sugar acids, sugar alcohols, and glycosides, using the API 50 CHL test. Among the tested carbohydrates, all *Leuc. carnosum* isolates (*n* = 51) were found to utilize four specific carbohydrates, namely D-glucose, D-fructose, D-sucrose, and esculin (Figure 3a). Most of the isolates also utilized D-ribose, N-acetyl-glucosamine, D-trehalose, D-turanose, and gluconate, while a lower utilization ability was observed for methyl alpha D glucopyranoside, D-mannose, gentiobiose, cellobiose, and maltose. D-galactose, D-melibiose, D-melezitose, D-raffinose, glycogen, mannitol, salicin, amygdalin, and arbutin were found to be utilized specifically by certain isolates. Based on the carbohydrate utilization patterns of the tested *Leuc. carnosum* PFGE types were categorized into 13 different biochemical profiles (BP). Among these profiles, the PFGE types from the raw meat showed the most distinctive biochemical patterns compared to other *Leuc. carnosum* PFGE types. For example, the PFGE type Leuc 6 from raw meat showed the highest carbohydrate utilization of 37.7% (BP 13). The PFGE type Leuc 2 isolated from raw meat showed carbohydrate utilization of 20.4% (BP 5), which included fermentation of D-mannose, D-galactose, D-maltose, D-melezitose, and D-raffinose, which was not observed among the isolates from cooked ham. The carbohydrate utilization of PFGE type Leuc 2 isolates from cooked ham ranged between 14.3% (BP 1), 16.3% (BP 2), 18.4% (BP 4 and BP 6), and up to 20.4% (BP 7). The BP 1 and BP 2 profiles were shared by two PFGE types (Leuc 2 and Leuc 3), while the most frequently observed biochemical profile (BP 4) in cooked ham was shared by three PFGE types (Leuc 2, Leuc 3, and Leuc 4). The PFGE types (Leuc 7, Leuc 8, and Leuc 11) from the FPE showed three biochemical profiles (BP8, BP9, and BP10). The BP9 and BP10 were shared between different PFGE types (Leuc 8, Leuc 9, Leuc 9-ST, Leuc 1, Leuc 1-ST, Leuc 10, and Leuc 11) isolated from FPE and cooked ham. We observed no difference in carbohydrate utilization among isolates with identical PFGE types that were isolated at the beginning, during, and at the end of cooked ham storage.

All *Lb. sakei* isolates (*n* = 48) were found to utilize five carbohydrates, namely D-glucose, D-fructose, N-acetyl glucosamine, D-mannose, and D-galactose (Figure 3b). Most of the *Lb. sakei* isolates utilized D-ribose, L-arabinose, D-sucrose, D-trehalose, gentibiose, D-cellobiose, D-melibiose, gluconate, and esculin. However, D-xylose, D-turanose, D-maltose, D-lactose, salicin, amygdalin, and arbutin were utilized specifically by certain isolates. Based on the carbohydrate utilization patterns of the tested *Lb. sakei* PFGE types, they were categorized into 14 distinct BP. Two PFGE types (Sakei 7 and Sakei 10) that shared BP 7 utilized 26.5% of carbohydrates when isolated from raw meat, FPE, and cooked ham. Other PFGE types (Sakei 3 and Sakei 4) isolated from raw meat, FPE, and cooked ham displayed slight differences in carbohydrate utilization percentages. The PFGE type Sakei 4 carbohydrate utilization ranged between 22.4% (BP 1), 24.5% (BP 3), and 26.5% (BP 8). The PFGE type Sakei 3 was represented by BP 6 and BP 5 when isolated from the raw meat, FPE, and cooked ham. The BP6 utilized 24.5% of carbohydrates, while BP 5 lacked salicin utilization and fermented 22.4% of carbohydrates. The most common biochemical patterns in cooked ham were BP 3 and BP 5, while sporadically isolated PFGE types (Sakei 6, Sakei 1, and Sakei 5) in cooked ham utilized 26.5% (BP9), 28.6% (BP10 and BP11) of carbohydrates. The same biochemical profile was not shared by different PFGE types in cooked ham, as observed with *Leuc. carnosum* PFGE types. We observed no difference in carbohydrate utilization among isolates with identical PFGE types that were isolated at the beginning, during, and at the end of cooked ham storage. The highest utilization ability of 32.7% carbohydrates (BP 14) was observed for PFGE type Sakei 9, isolated from brine (BP 14). Additionally, other PFGE types from FPE, including Sakei 4-ST, Sakei 11, and Sakei 8, isolated from brine, tumbler after sanitation, and cutting board, utilized 22.4% (BP 2), 28.6% (BP 12), and 30.6% (BP 13) carbohydrates, respectively.

The *Leuc. carnosum* PFGE types had positive acid and alkaline phosphatase enzymatic activity (Appendix A). Furthermore, naphthol-AS-BI-phosphohydrolase activity was present in all PFGE types except for Leuc 11 and Leuc 12. The *Lb. sakei* PFGE types had positive leucine and valine arylamidase enzymatic activity. The highest enzymatic activity was observed in PFGE type Sakei 11, identified after tumbler sanitation (Appendix A).

## 4. Discussion

The marketing of ready-to-eat convenience products, such as cooked ham, is increasing due to consumer demand for less processed foods and high organoleptic quality [1,30,31]. However, cooked ham is a perishable meat product with pH values ranging from 5.5 to 6.5, water activity (aw) between 0.95 and 0.99, and readily available nutrients such as glucose, ribose, amino acids, and nucleosides. These characteristics make it an ideal growth medium for a wide range of microorganisms originating from raw meat, other ingredients, or the food processing environment (FPE), which can potentially compromise the safety and shelf life of the final product [10,32]. Vacuum packaging and cold storage are commonly used to prevent the growth of spoilage microorganisms. However, these measures often result in selective pressure towards psychrotrophic and strictly or facultatively anaerobic microbes, such as LAB [10].

We used a maximum acceptable limit of 7.4 log_10_ AMC/LAB CFU/g for the end of shelf life in cooked ham, in accordance with the AMA quality seal requirements in Austria [21]. The aim of the study was the determination of the timepoint when the maximum acceptable limit during cold storage of cooked ham was exceeded. Other questions included the effect of the microbial load and composition of the raw meat on the microbial limit of the cooked ham and the effect of the hygiene conditions in the pre- and post-cooking areas of the FPE on the exceeding of the shelf life of the cooked ham. The microbial growth dynamics in nine different lots of cooked ham varied, with initially low LAB populations. The LAB population count in cooked ham exceeded the maximum acceptable microbial limit on different days, with a minimum of 11 and a maximum of 20 days of storage. Even when the raw meat was treated with brine cultures to extend the shelf life of the cooked ham, the maximum acceptable limit was exceeded during storage of the cooked ham on days 11 and 20. These observations are not surprising, as the shelf life of vacuum-packed or MAP-packed processed meat products, including cooked ham, is generally dictated by LAB growth [1,10]. Moreover, previous studies also reported rapid LAB growth from initially low LAB levels in freshly packed cooked ham [5,30,33]. The intrinsic characteristics of cooked ham, such as water activity, sugar-enriched preparations, and the presence of sodium chloride and sodium nitrite, provided a selective advantage for the growth of strict and facultative anaerobic LAB [7,8,12,30]. The observation of a plateau in LAB growth when the limit of 7.4 log10 CFU/g was exceeded could be attributed to acid production and nutrient depletion, as previously reported when LAB counts reached 8.0 or 9.0 log_10_ CFU/g [5,33]. Several other studies observed LAB growth above 7.0 log_10_ CFU/g in cooked ham also reported acid flavor, slime and gas production, off-odors, and cooked ham discoloration [5,18,34]. However, in contrast to these studies, we did not observe any sensory alterations in cooked ham, except for the decrease in pH, which was only measured during the storage of lots 8 and 9. These observations suggest that the spoilage process is not solely caused by the microbial count but also by the accumulation of metabolic byproducts from specific microorganisms [35]. Although other LAB bacteria (*L. mesenteroides*, *Carnobacterium divergens*, *Leuconostoc gelidum* subsp. *gasicomitatum*) and non-LAB bacteria (*Pseudomonas*, *Kocuria*, *Corynebacterium*, *Bacillus*, *Staphylococcus*, *Enterococcus*) were isolated at the beginning of storage, they were outcompeted by the rapid growth of *Leuc. carnosum* and *Lb. sakei.* However, a slimy surface texture was observed during storage of cooked ham lot 9. The observed spoilage effects could be attributed to the isolation of spoilage organisms, such as *L. gelidum* subsp. *gasicomitatum*, *L. curvatus*, *C. divergens*, which are known to form slime and produce acidic or buttery off-flavors [10]. Both species were identified as part of a raw meat psychrotrophic microbiota identified in meat during the tumbling process, following their isolation in cooked ham [12]. It was surprising to observe the absence of microbial growth during the 20 days of storage of cooked ham lot 6. This observation is consistent with the results reported by Zagdoun et al. 2021 [9], where AMC and LAB counts showed a lag phase around 22 and 28 days and an exponential phase around 40 days of storage. The absence of microbial growth observed during the storage of lot 6 may be explained by the limited selectivity of the ISO method, particularly with regard to psychotropic bacteria, which are typically underestimated [9,13]. Consistent with our findings, Zagdoun et al. [9] also reported fluctuations in LAB counts during the storage period, along with variability in the initial concentration at the beginning of the storage. This suggests that the observed variability in LAB growth among different lots of cooked ham may be attributed to cross-contamination events occurring under the conditions of the processing facility [9].

In determining the final microbiota of cooked ham, the raw meat, the processing environment, and the hygiene conditions throughout the processing line play a crucial role. In addition to investigating the AMC/LAB level during cooked ham storage, our study focused on evaluating the microbial load and composition of the raw meat input, the hygiene conditions in the FPE, and in particular, the isolation of LAB, which dominates under psychrotrophic conditions during cooked ham storage.

Therefore, after assessing the initial microbial status of the three cooked ham lots, we also conducted microbiological investigation of the raw material in subsequent production lots. We did not observe any association between the microbial load of the raw meat and how fast the microbial limit was exceeded during cooked ham storage due to the small amount of analyzed raw meat samples. A limitation of the study is the small number of raw meat samples analyzed. However, the isolation of LAB from raw meat and FPE and subsequent identification of the isolates by molecular subtyping confirmed the contamination of the final product by raw meat. Samelis et al. [20] reported a 20% presence of typical *L. carnosum* strains on raw pork meat used for cooked ham production. Other studies have reported recontamination of products in the post-cooking area prior to packaging with about 0.5–2.0 log_10_ CFU/g of mainly LAB [18]. The characterization of the isolates by molecular typing indicated that the contamination of the final product originated from the raw meat and the post-cooking area. The most frequently isolated strains included *Leuconostoc carnosum* (Leuc) PFGE types Leuc 2, Leuc 3, and *Latilactobacillus sakei* (Sakei) PFGE types Sakei 3 and Sakei 4, which were both present in raw meat and the post-cooking area of the FPE. Prior studies have addressed the impact of raw meat and FPE on the composition of the final microbiota in cooked ham mainly by 16S rRNA gene profiling and by culture-dependent methods [9,11,12,16,19,20]. However, the present study provides confirmation of these observations at the strain level.

In eight out of nine cooked ham lots tested, there was no microbial growth at the beginning of storage (day 0), indicating appropriate hygienic and processing conditions, which were supported by the low microbial counts observed on the surfaces in the pre- and post-cooking areas of the FPE as proposed by Garriga et al. [30]. These observations indicated that even if the microbial count in FPE is low, recontamination of the end product with a small fraction of specific LAB strains can reduce the shelf life of cooked ham. In fact, both LAB species, *Leuc. carnosum* and *Lb. sakei*, were identified in the tumbler after disinfection. The meat production plant uses a combined alkaline detergent-sanitizer (DS), which can eliminate most microorganisms. However, it has been shown that this type of disinfectant is not sufficient to remove LAB [20]. Furthermore, the absence of LAB growth at the start of storage can be attributed to sublethal damage during the cooking step at 72 °C for 60 min. However, with prolonged storage, the sublethally damaged LAB resuscitated, leading to the microbial limit being exceeded. Veselá et al. [12] demonstrated a similar phenomenon, where sublethally damaged LAB were able to resuscitate after cooking at a core temperature of 72 °C for 10 min during storage of cooked ham.

Previous studies have reported variability in biochemical characteristics among *Leuconostoc* sp. and *Latilactobacillus* sp. strains isolated from various sources, including dairy-related products, modified atmosphere packaged sausage, and cooked ham [10,36,37]. These findings align with our observations. In addition, biochemical characterization of frequently isolated strains from cooked ham, raw meat, and FPE revealed specific adaptations in carbohydrate utilization, reflecting their adaptation to the source of isolation. For instance, the PFGE type Leuc 2 from raw meat indicated a distinct carbohydrate utilization pattern by fermenting D-mannose, D-galactose, D-maltose, D-melezitose, and D-raffinose, which was not observed among the isolates from cooked ham. Similarly, *Lb. sakei* PFGE types Sakei 3 and Sakei 4 from raw meat, FPE, and cooked ham showed slight differences in carbohydrate utilization patterns, including D-melibiose and salicin, depending on the isolation source. These observations emphasize the different fermentative capabilities among the strains of the same species. Therefore, the growth capabilities and dynamics in the meat matrix, competitiveness against other bacteria, and production of spoilage-related molecules of *Leuc. carnosum* and *Lb. sakei* strains identified in the present study need to be investigated [10]. Some *Leuc. carnosum* and *Lb. sakei* originating from unspoiled samples have already been considered potential bioprotective cultures in meat-based products in other studies [1,20,38,39].

Overall, the data demonstrate that the presence of LAB in raw meat and FPE influenced the exceeding of the microbial limit during the cooked ham storage without causing sensory defects. Our observations are in line with the recent observations of Alessandria et al. [1], who also highlighted the discrepancy between microbial counts and sensory defects in cooked ham at the end of shelf life. They observed that some cooked hams with higher microbial counts did not exhibit sensory defects. Furthermore, some samples showed similar independent microbial counts in spoiled and unspoiled cooked ham samples. Similarly, the microbiota analysis conducted by Raimondi et al. [9] in the cooked ham at the end of its shelf life did not identify any specific taxon association with cooked ham spoilage. These findings suggest that the spoilage effects may be linked to specific strain characteristics within the same species or possibly to subdominant bacterial groups.

The present study emphasized the challenges of controlling LAB growth in vacuum-packed cooked ham despite high hygiene standards and processing conditions. In order to ensure food safety, it is important to consider that the presence of low LAB numbers in raw meat and FPE can limit the shelf life. Furthermore, future research should focus on understanding the spoilage potential of specific strains of LAB that are associated with sensory alterations in cooked ham. Instead of relying solely on a strict limit of 7.4 log_10_ CFU/g to determine the end of shelf life, it is important to consider the specific characteristics and behaviors of LAB strains that contribute to spoilage in cooked ham.

## 5. Conclusions

There is an increasing consumer demand for cooked ham with an extended shelf life. The presence of lactic acid bacteria during cooked ham storage has a controversial role due to discrepancies between the maximum acceptable microbial load and sensory deficiencies during cooked ham storage. The analysis of a large number of samples during the storage of cooked ham demonstrated that lactic acid bacteria, *Leuc. carnosum* and *Lb. sakei*, were responsible for exceeding the microbial limit for the end of shelf life without observed sensory deficiencies after 20 and 33 days of storage. Furthermore, the presence of the most frequently identified *Leuc. carnosum* PGFE types Leuc 2 and *Lb. sakei* PFGE types Sakei 3 and Sakei 4 in cooked ham was associated with raw meat and recontamination events in the post-cooking area of the food processing environment, including the unwrapping and cutting of cooked ham. While controlling the growth of psychrotrophic facultative strict and facultative anaerobic lactic acid bacteria in cooked ham remains a challenge, it may be advisable to focus on the identification of specific spoilage organisms instead of relying solely on a strict microbial limit to determine the end of shelf life in the cooked ham.

## Figures and Tables

**Figure 1 foods-12-02475-f001:**
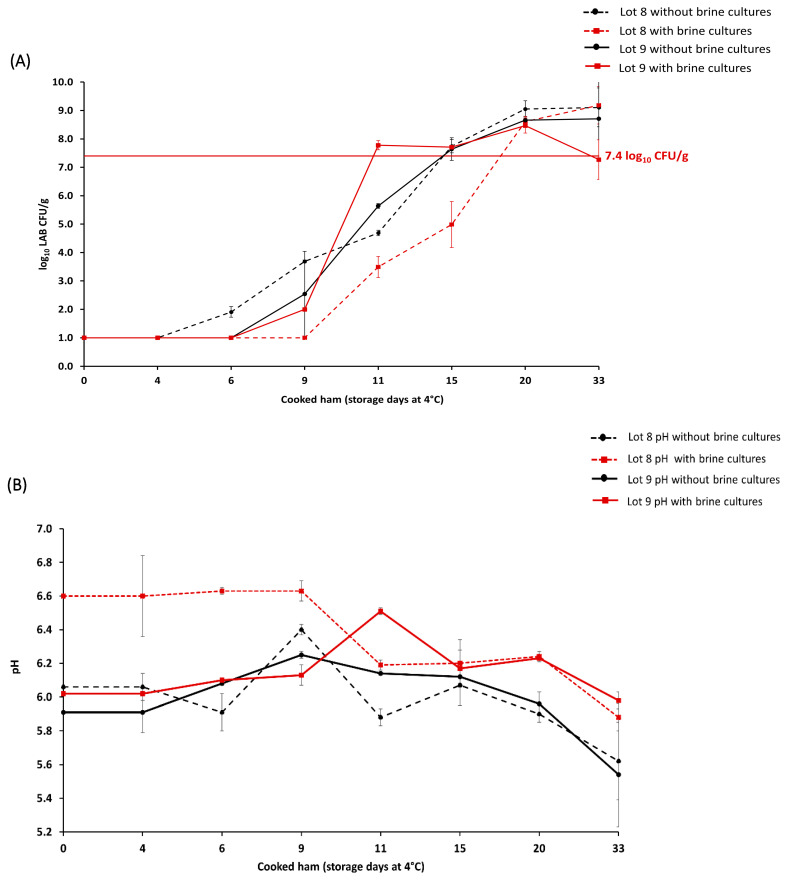
Average lactic acid bacteria (LAB) count (log_10_ CFU/g) in cooked ham lots 8 and 9 with (*n* = *32*) and without (*n* = 32) brine cultures until day 33 of storage (**A**). The maximum acceptable microbial limit of ≥7.4 LAB log_10_ CFU/g during the cooked ham storage is marked with a horizontal, full red line. pH in cooked ham lots 8 and 9 measured until the day 33 of storage (**B**). The representation of cooked ham without brine cultures (●) is denoted in black, while samples with brine cultures (■) are shown in red.

**Figure 2 foods-12-02475-f002:**
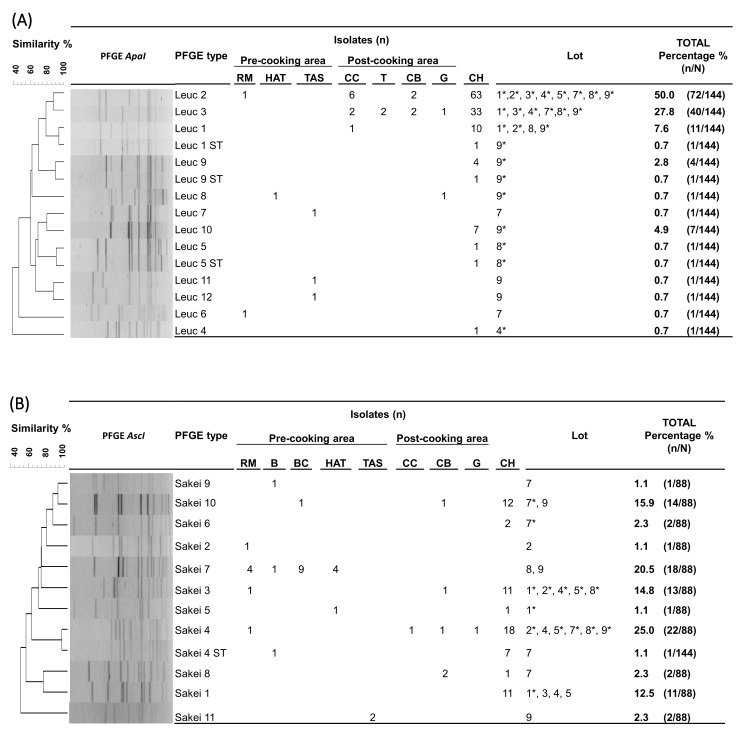
Unweighted pair group method with arithmetic mean (UPGMA) dendrogram with dice coefficient and 1.5% position tolerance among *XbaI* and *AscI* pulsed-field gel electrophoresis (PFGE) patterns of *Leuconostoc (Leuc.) carnosum* (**A**) and *Latilactobacillus (Lb.) sakei* (**B**) isolates from raw meat, food processing environments, and cooked ham. Abbreviations: B: brine; BC: brine cultures; CB: cutting board; CH: cooked ham G: gloves during packaging of the end product; HAT: ham after tumbling; CC: cellulose casing; RM: raw meat; RMDB: raw meat delivery box; T: trolley; TAS: tumbler after sanitation; *: represents the isolation of the PFGE type from cooked ham.

**Figure 3 foods-12-02475-f003:**
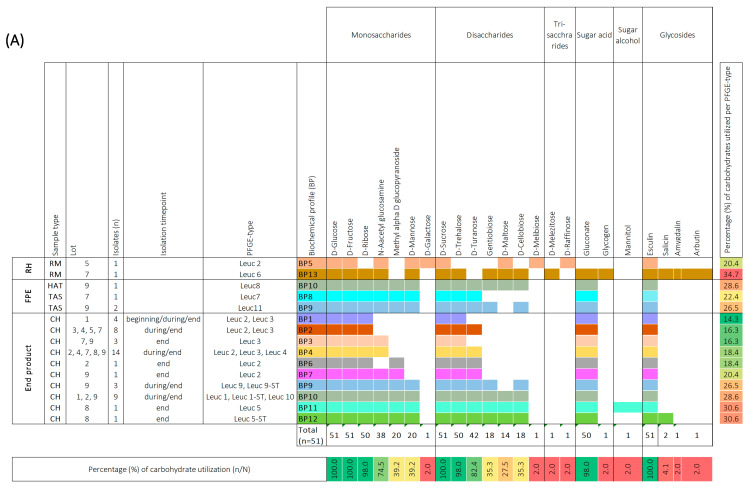
Biochemical profiles were generated based on the 49-carbohydrate utilization ability of *Leuconostoc (Leuc.) carnosum* (*n* = 51) (**A**) and *Latilactobacillus (Lb.) sakei* (n = 48) (**B**) isolates. All carbohydrate substrates (API 50 CHL) were categorized into monosaccharides, disaccharides, trisaccharides, sugar acids, sugar alcohols, and glycosides. The isolate’s ability to utilize a carbohydrate is represented as a positive (coloration) or negative (white color) reaction. The carbohydrates that were not utilized by a single isolate are not represented in the figure. The isolates were categorized according to sample type into raw meat (RM), food processing environment (FPE), and end product. The *Leuc. carnosum* and *Lb. sakei* PFGE types were color-coded representing different biochemical fermentation profiles and carbohydrate utilization patterns. The numbers on the right represent the percentage of carbohydrates utilized per isolate, while the numbers on the bottom represent the percentage of a single carbohydrate per studied group of isolates. Gradient scale: 0% (red)—100% (green). Abbreviations: B: brine; BC: brine cultures; CB: cutting board; CH: cooked ham G: gloves during packaging of the end product; HAT: ham after tumbling; CC: cellulose casing; RM: raw meat; RMDB: raw meat delivery box; T: trolley; TAS: tumbler after sanitation.

**Table 1 foods-12-02475-t001:** Overview of raw meat (RM; *n* = 12), food processing environment (FPE; *n* = 34) and cooked ham (CH; *n* = 122) sampling. Abbreviations: B: brine; BC: brine cultures; CB: cutting board; CC: cellulose casing; CH: cooked ham; G: gloves during packaging of the end product; HAT: ham after tumbling; RM: raw meat; RMDB: raw meat delivery box; T: trolley; TAS: tumbler after sanitation.

Lot	RM	BC	FPE									CH								
			Pre-Cooking					Post-Cooking				Storage Days								
	*n* = 12		*n* = 14					*n* = 20												*n* = 122
			RMDB	B	BC	HAT	TAS	CC	T	CB	G	0	4	6	9	11	15	20	33	
1		*no*										2				2		2		6
2		*no*										2				2		2		6
3		*no*										2				2		2		6
4	2	*no*										2	2	2		2				8
5	2	*no*										2	2		2	2				8
6	2	*no*										2		2		2	2	2		10
7	2	*no*		1			1	1	1	1	1	2	2	2	2	2	2	2		14
8	2	*no*		1				1	1	1	1	2	2	2	2	2	2	2	2	16
		*yes*		1	1			1	1	1	1	2	2	2	2	2	2	2	2	16
9	2	*no*	1	1		2	1	1	1	1	1	2	2	2	2	2	2	2	2	16
		*yes*		1	1	2		1	1	1	1	2	2	2	2	2	2	2	2	16

**Table 2 foods-12-02475-t002:** Average aerobic mesophilic count (AMC), lactic acid bacteria (LAB), *Enterobacteriaceae* (EB) and *Pseudomonadaceae* (PS) counts are provided as mean values (log_10_ CFU/g) and standard deviations in raw meat (RM) (*n* = 12) used for cooked ham production (lots 4–9). The presence, absence, or partial presence of specific microorganisms in the samples is indicated as follows: (+) presence in two biological replicates; (+/−) presence in one biological replicate; or (−) absence from two biological replicates *of Leuconostoc (Leuc.) carnosum*, *Latilactobacillus (Lb.) sakei* and other bacteria identified with partial 16S rRNA gene sequencing.

	Mean log_10_ CFU/g					
Lot	AMC	LAB	EB	PS	*Leuc. carnosum*	*Lb. sakei*
**4**	5.4 ± 0.75	3.2 ± 0.51	4.0 ± 0.62	<1.0	−	+
**5**	4.8 ± 0.17	3.4 ± 0.30	1.1 ± 1.29	2.3 ± 0.62	+/−	−
**6**	4.1 ± 0.27	2.6 ± 0.28	<1.0	1.7 ± 0.40	−	−
**7**	5.7 ± 0.08	4.0 ± 0.37	4.1 ± 0.44	4.6 ± 0.42	+/−	−
**8**	3.4 ± 0.05	3.0 ± 0.21	<1.0	<1.0	−	+
**9**	5.9 ± 0.43	5.8 ± 0.07	4.7 ± 0.13	4.7 ± 0.33	−	−
Percentage (%) (n/N)					16.7 (2/12)	33.3 (4/12)

**Table 3 foods-12-02475-t003:** Average aerobic mesophilic count (AMC) and lactic acid bacteria (LAB) of cooked ham samples (lots 1–9, *n* = 90) are provided as mean values and standard deviations. The presence, absence, or partial presence of specific microorganisms in the samples is indicated as follows: (+) presence in two biological replicates; (+/−) presence in one biological replicate; or (−) absence from two biological replicates. *Leuconostoc (Leuc.) carnosum*, *Latilactobacillus (Lb.) sakei*, other LAB (e.g., *Latilactobacillus graminis*, *Leuconostoc mesenteroides*, *Weissella viridescens*), and non-LAB (e.g., genus *Bacillus*, *Corynebacterium*, *Enterococcus*, *Pseudomonas*, *or Staphylococcus*) were identified with partial 16S rRNA gene sequencing.

Lot	Storage Days	Mean Log_10_ CFU/g		*Leuc. carnosum*	*Lb. sakei*	Other	Non-LAB
		AMC	LAB			LAB	Bacteria
1	0	3.5 ± 0.01	3.5 ± 0.00	+	+	−	+
	11	7.8 ± 0.02	7.8 ± 0.06	+	+	−	−
	20	8.3 ± 0.01	8.3 ± 0.01	+	+	−	−
2	0	<1.0	<1.0	−	−	−	+
	11	7.3 ± 0.32	7.3 ± 0.35	+	+	−	−
	20	7.9 ± 0.04	7.8 ± 0.03	+	+	+/−	−
3	0	<1.00	<1.0	−	−	−	+
	11	6.8 ± 0.02	6.8 ± 0.07	+	−	+/−	−
	20	8.3 ± 0.06	8.3 ± 0.08	+	+	−	−
4	0	1.3 ± 0.43	<1.0	−	−	−	+
	4	3.1 ± 0.47	2.9 ± 0.20	+	−	+	+/−
	6	3.8 ± 0.23	3.7 ± 0.21	+	+	−	−
	11	7.6 ± 0.16	7.4 ± 0.42	+	+	−	−
5	0	<1.0	<1.0	−	−	−	−
	4	<1.0	<1.0	−	−	−	−
	9	6.9 ± 0.02	7.0 ± 0.33	+	+	−	−
	11	8.0 ± 0.27	7.7 ± 0.03	+	+	−	−
6	0	<1.0	1.0 ± 0.03	−	−	−	−
	6	<1.0	<1.0	−	−	−	−
	11	<1.0	<1.0	−	−	−	−
	15	<1.0	<1.0	−	−	−	−
	20	1.7 ± 0.57	<1.0	−	−	−	−
7	0	<1.0	<1.0	−	−	−	−
	4	1.7 ± 0.20	1.9 ± 0.21	+	−	−	−
	6	4.3 ± 0.34	4.2 ± 0.21	+	+	−	−
	9	4.6 ± 0.20	5.0 ± 0.39	+	+	−	−
	11	7.4 ± 0.01	7.2 ± 0.26	+	+	−	−
	15	7.9 ± 0.59	8.3 ± 0.29	+	+	−	−
	20	8.2 ± 0.57	8.4 ± 0.04	+	+	+/−	−
8	0	<1.0	<1.0	−	−	−	−
	4	<1.0	<1.0	−	−	−	−
	6	2.1 ± 0.27	1.9 ± 0.19	+	−	−	−
	9	3.8 ± 0.14	3.7 ± 0.05	+	−	−	−
	11	4.6 ± 0.09	4.7 ± 0.09	+	−	−	−
	15	7.7 ± 0.05	7.7 ± 0.23	+	−	−	−
	20	9.2 ± 0.31	9.1 ± 0.29	+	−	−	−
	33	9.2 ± 0.64	9.1 ± 0.68	+	−	−	−
9	0	<1.0	<1.0	−	−	−	−
	4	1.5 ± 0.00	<1.0	−	−	−	+/−
	6	<1.0	<1.0	−	−	−	−
	9	1.4 ± 0.12	2.5 ± 1.50	−	−	−	+/−
	11	6.3 ± 0.82	5.6 ± 0.08	+	+	+/−	−
	15	7.5 ± 0.27	7.6 ± 0.40	+	+	−	−
	20	8.7 ± 0.08	8.7 ± 0.06	+	−	+/−	+/−
	33	7.6 ± 0.42	7.3 ± 0.70	+	−	−	−
Percentage (%) (n/N)				62.2 (56/90)	37.8 (34/90)	7.8 (7/90)	13.3 (12/90)

**Table 4 foods-12-02475-t004:** Average aerobic mesophilic count (AMC) and lactic acid bacteria (LAB) counts of the food processing environment (FPE) (lots 7–9; *n* = 34). The dashed line represents the distinction between pre- and post-cooking areas. The presence (+) or absence (-) of *Leuconostoc (Leuc.) carnosum*, *Latilactobacillus (Lb.) sakei*, and other bacteria (e.g., genus *Brochotrix*, *Carnobacterium Pseudomonas*, or *Psychrobacter*) were identified with partial 16S rRNA sequencing. Abbreviations: B: brine; BC: brine cultures; CB: cutting board; G: gloves during packaging of the end product; HAT: ham after tumbling; CC: cellulose casing; RM: raw meat; NA, not applicable; RMDB: raw meat delivery box; T: trolley; TAS: tumbler after sanitation.

FPE Sampling Area		AMC			LAB			*Leuc.* *carnosum*		Non-LAB Bacteria Number of Isolates
	Sample	Log_10_ CFU/cm^2^			Log_10_ CFU/cm^2^				*Lb.* *sakei*	
	(log_10_ CFU/ mL or cm^2^)	Lot 7	Lot 8	Lot 9	Lot 7	Lot 8	Lot 9			
	RMDB	NA	NA	3.1	NA	NA	3.1	−	−	4
	B	3.3	1.2–3.9	3.2–3.8	2.9	<1.0–3.7	2.7–3.9	−	+	14
	BC	NA	10.5	13.1	NA	10.8	13.1	−	+	2
Pre-cooking	TAS	1.9	NA	3.0	1.9	NA	3.0	+	+	11
	HAT	NA	NA	4.6–6.6	NA	NA	3.9–6.5	+	+	10
	CC	1.7	1.2–1.5	1.3–1.4	<1.0	<1.0–1.1	<1.0–1.2	+	+	16
Post-cooking	T	<1.0	<1.0–1.2	<1.0–1.1	<1.0	<1.0	<1.0	+	−	11
	CB	1.4	1.7–2.4	1.9–2.4	1.5	<1.0–1.9	<1.7–2.3	+	+	13
	G	<1.0	1.1–1.7	1.0–1.4	<1.0	<1.0–1.2	<1.0	+	+	10
	Percentage (%) (n/N)							35.3 (12/34)	35.3 (12/34)	

## Data Availability

Data is contained within the article or Appendix A.

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
