# Peer review of "Characterization of Leuconostoc carnosum and Latilactobacillus sakei during Cooked Pork Ham Processing"

_foods, 2023, doi:10.3390/foods12132475_

Round 1

Reviewer 1 Report

The paper is interesting and many data are reported. It is general well written, but some grammar errors (see specific comment below) and inaccuracies are present. For example, the sampling approach is sometimes confusing: see for example lot 8, or the reason why only the raw material of some samples was taken. Then, I do not understand why the authors report in tables the different suppliers, if then this variable is never considered in the discussion of results. I would rather remove this information. In addition, in my opinion to many tables (even if supplementary) are present, and for example tab S4 could be removed since it gives few information already reported in the text. I would consider also to use part of the data for a second paper: for example, from paragraph 3.5 onwards, since the strain characterization could stand almost alone in another manuscript (if some other studies on the specific strains are in progress), but I refer this decision to the editor of the journal.

Finally, the discussion is mainly a summary of the results, and the “real” discussion of the data in comparison with the available literature is quite limited.

Specific comments:

Why the discussion has a new line numbering?

Line 16: I would say “its composition in nine…using plate counting”

Line 19: Add” the strains …” at the beginning of the sentence

Line 21: I would add “isolated” from raw material, or something similar

Line 23 and 35: remove comma

Line 37: Psychrotrophic instead of psychotropic

Line 52: …can allow LAB to grow up to 8.0 log…

Line 61: please define AMC

Line 69 and throughout the whole manuscript: Leuconostoc and Latilactobacillus should be abbreviate differently (not L. for both). The same in Tab S1.

Lines 98-102: what about lot 8? It is not mentioned here but only below for the final products.

Table 1: what is the meaning of the number 2 after storage days (right side of the table)?

Line 131: and pH measure

Lines 208- 211: the verb is missing.

Line 267-269: I would add this information above, in order to make a single sentence

Line 288: lot 4 to 9 (and the same in the tables/figure), while in M&M the authors say that RM was samples in lot from 4 to 7 (line 99). Please correct/clarify.

Line 300: were isolated (yeastS)

Lines 306-307: what about lot 7? In fig. 1 the cell load is higher than 1 (it seems about 1.8), but in table 3 the value reported is <1.

Line 333: horizontal full red line (not vertical)

Lines 343-361: why this case the microbial limit considered LAB counts, while before (fig 1) it refers to AMC?

Line 349: the limit is exceed before 20 days (17-18?)

I would remove table S4, writing “data not shown”. There are two many tables (even if supplementary)

Caption table 4: some abbreviations are reported in the caption ma those samples are no present in the table (CH, RM). Please correct.

Line 380: remove microbial after brine

DISCUSSION

Line 8: I would say “selective pression”

Line 9: Psychrotrophic instead of psychotropic

Some minor errors are present (see above)

Author Response

AUTHORS (AU): Thank you very much for your thorough review of the manuscript. We have now revised the manuscript, according to your kind suggestions. The changed sections have been highlighted in yellow.

Academic Editor's suggestion: I prefer not to split the work.

Specific comments:

Why the discussion has a new line numbering? AU: Thank you for your comment, has been changed.

Line 16: I would say “its composition in nine…using plate counting” AU: thank you, was changed.

Line 19: Add” the strains …” at the beginning of the sentence AU: was added.

Line 21: I would add “isolated” from raw material, or something similar AU: was added.

Line 23 and 35: remove comma AU: was removed.

Line 37: Psychrotrophic instead of psychotropic AU: thank you, was changed.

Line 52: …can allow LAB to grow up to 8.0 log…AU: was changed

Line 61: please define AMC AU: the abbreviation was written out

Line 69 and throughout the whole manuscript: Leuconostoc and Latilactobacillus should be abbreviate differently (not L. for both). The same in Tab S1. AU: thank you very much Latilactobacillus (Lb.) sakei and Leuconostoc (Leuc.) carnosum were abbreviated according to your suggestion

Lines 98-102: what about lot 8? It is not mentioned here but only below for the final products.

 AU: thank you for your input. The section was modified according to the suggestion of other reviewers. Raw material was mentioned for lot 8 and 9 (now L132).

Table 1: what is the meaning of the number 2 after storage days (right side of the table)? AU: thank you was an error, was deleted.

Line 131: and pH measure AU: thank you was amended.

Lines 208- 211: the verb is missing. AU: thank you the sentences were modified.

Line 267-269: I would add this information above, in order to make a single sentence

AU: was changed according to your suggestion.

Line 288: lot 4 to 9 (and the same in the tables/figure), while in M&M the authors say that RM was samples in lot from 4 to 7 (line 99). Please correct/clarify.

AU: This stylistic ambiguity has been resolved in the newly revised Materials and Methods section. The raw material was tested as part of the investigation of batches 4-9. The testing of batches 8 and 9 using brine additives is described separately.

Line 300: were isolated (yeastS)

AU: was ammended

Lines 306-307: what about lot 7? In fig. 1 the cell load is higher than 1 (it seems about 1.8), but in table 3 the value reported is <1.

AU: figure 1 which included the AMC values has been deleted, the values are also listed correctly in table 3. Thank you for pointing out the error in lot 7 in the former figure 1 (AMC)

Line 333: horizontal full red line (not vertical)

AU: thank you was changed

Lines 343-361: why this case the microbial limit considered LAB counts, while before (fig 1) it refers to AMC?

AU: we have deleted figure 1 showing AMC values to make the dataset clearer and refer to the important information of the LAB limit.

Line 349: the limit is exceed before 20 days (17-18?)

AU: the limit was exceeded on day 20, it is a graphical representation problem

I would remove table S4, writing “data not shown”. There are two many tables (even if supplementary)

AU: We have kept Table S4 because we think it helps to better understand the dataset.

Caption table 4: some abbreviations are reported in the caption ma those samples are no present in the table (CH, RM). Please correct.

AU: thank you was corrected.

Line 380: remove microbial after brine

AU: thank you was corrected

DISCUSSION

Line 8: I would say “selective pression”

AU: we have checked the wording of selective pression and selective pressure; selective pressure is what we kept in mind

Line 9: Psychrotrophic instead of psychotropic AU: thank you was corrected.

Reviewer 2 Report

This manuscript presents in the changes of AMC and LAB during cooked pork ham processing. A lot of time was spent on the isolate collection and identification of microorganisms and molecular subtyping of Leuconostoc carnosum and Latilactobacillus sakei. It is a very difficult job. However, the sampling of the experimental is not a balanced experimental design. The supplier of hams and the number of days for storing experimental are not the same in every experiment. It is recommended to redesign the experiments to fill in the missing experimental points.

If it is not possible to complete the experiment, please give the reasons for adopting this experimental design in this study.

#1. When using abbreviations for the first time in an article, please specify the full name. For example, the AMA in P2 Line64 and the PFGE in P2 Line72. Please recheck the whole manuscript.

#2. In Table 1, the data of supplier C is missing in Lot 1-3, and the data of suppliers C and D are missing in Lot 4-6. Although it is stated that more raw meat was analyzed in the Lot 4-6. Please provide the reason for this unbalanced in the experimental design. In addition, there is no common rule for the number of sampling days for Lot 1-3 and Lot 4-6 of storage experiments. Please provide reasons for conducting the experiment in this way.

#3. In Table 1, there is only one group of experiments with microbiological analysis of RMDB in Lot8-9, please explain why the experiments are designed in this way.

#4. In P7 Line307-308, Lot 1 and 4 have higher raw bacteria counts, but the bacteria counts after storage are not the highest at the end of the storage experiment. Please discuss if there are other factors that affect the microbial changes in the storage process of hams. In addition, Lot 4 and Lot 6 are both the products of suppliers A, but Lot 6 was found to have the lowest microbial counts during the storage process. Please provide a reasonable explanation and supporting references.

#5. Please explain the meaning of NA in Table 4.

#6. In Table 4, Lot 9 had the highest microbial counts in the microorganism in environment, but hams of Lot 9 did not have the highest microbial counts in the storage period among these groups. Please provide possible explanations and supporting references.

#7. The bacteria count in the Pre-cooking group is higher than the Post-cooking group, which can be considered as a significant reduction of microorganisms in the sterilized ham. However, it is not clear from the experimental methods whether sampling is done in ham production when monitoring microbial levels in the environment. Because it is mentioned P3 Line112-113 that “Swab samples were collected from FPE along the processing line during the production of lot 7 to 9.” This description appears to sampling during the production process. The microbial counts detected in this way should come from the microorganisms contained in the ham itself. But it is mentioned P8 Line75 that “L. carnosum and L. sakei were isolated from the tumbler after cleaning with a combined alkaline detergent-sanitizer (DS)”. This description appears to be a post-production cleaning followed by sampling to verify the number of microorganisms in the environment that may be causing contamination. Please explain in detail the time point of sampling, and explain for these two parts of manuscript.

#8. The bacteria count in the raw meat of Lot6 in Table 2 was not the lowest among all groups. After sterilization, it can be seen in Table 3 that the bacteria count in lot 6 was not the lowest at first, but with increasing storage time, the microbial count in lot 6 decreased and remained very low. Please provide a reasonable explanation and supporting references.

#9. In P3 Line 65-67, there is mention of the many different spoilage phenomena caused by ham during storage. “(i.e. greenish colouring, discoloration, off-odor, slime production, milky exudates, and gas production)”. In P9 Line 122-123, it mentioned that “in combination with routine sensory evaluation, as these are crucial for determining the end of shelf-life”. However, sensory evaluation is generally used to assess changes in the strength of the sensory characteristics of the product during storage, rather than to assess the undesirable flavors that occur when the ham becomes rancid. It is suggested to revise these sections.

The quality of English Language is moderate, it is recommended to make appropriate corrections.

Author Response

AUTHORS (AU): Thank you very much for your thorough review of the manuscript. We have now revised the manuscript, according to your kind suggestions. The changed sections have been highlighted in yellow.

AU: This study was developed in close collaboration with a meat processing company. The slaughterhouses were not selected for the study, but are the actual main suppliers of the processing company. Raw materials are offered by many small and medium-sized suppliers in Austria. In reality, there is not a large choice of raw materials. As we do not mention and interpret the influence of slaughterhouses on raw material quality in this study, this information has been removed from the manuscript to provide more clarity regarding LABs. Ham production was not carried out on a laboratory scale, but on the processing facility under real conditions, in order to provide the collaborators with a realistic picture of the prevalence and potential variation of LABs during storage. It was also important to characterize the isolates to determine their potential spoilage characteristics.

#1. When using abbreviations for the first time in an article, please specify the full name. For example, the AMA in P2 Line64 and the PFGE in P2 Line72. Please recheck the whole manuscript. 

AU: Thank you for your comment. We have checked all abbreviations throughout the manuscript.

#2. In Table 1, the data of supplier C is missing in Lot 1-3, and the data of suppliers C and D are missing in Lot 4-6. Although it is stated that more raw meat was analyzed in the Lot 4-6. Please provide the reason for this unbalanced in the experimental design. In addition, there is no common rule for the number of sampling days for Lot 1-3 and Lot 4-6 of storage experiments. Please provide reasons for conducting the experiment in this way.

AU: thank you for your careful review. We have now removed the slaughterhouse information as it is not relevant to this study. We have modified the section 2.1 and 2.2 in materials and methods.

The study is divided into three empirical phases, resulting from the collaboration with the meat processing company. the first phase was to determine whether hams had elevated AMC/LAB values at the end of the shelf life and which microbes were responsible. the second phase also included the raw materials and whether these microbes were detectable after heating and the microbial diversity. the third phase investigated the effect of brine additives. The days of investigation included in the study were dependent on the actual production process of the ham in the production plant and therefore could not be equated in all studies, as everything happened dependent on the real production processes.

#3. In Table 1, there is only one group of experiments with microbiological analysis of RMDB in Lot 8-9, please explain why the experiments are designed in this way.

AU: In close cooperation with the meat processing company, the question was asked whether brine additives would stabilize the process and also contribute to the suppression, displacement of LAB during production. The first experiment showed that this brine additive mixture had no effect on the suppression of LAB after cooking the ham. Therefore, no further experiments were necessary. One of the main objectives of this study was to characterize the LAB species present in this production line.

#4. In P7 Line307-308, Lot 1 and 4 have higher raw bacteria counts, but the bacteria counts after storage are not the highest at the end of the storage experiment. Please discuss if there are other factors that affect the microbial changes in the storage process of hams. In addition, Lot 4 and Lot 6 are both the products of suppliers A, but Lot 6 was found to have the lowest microbial counts during the storage process. Please provide a reasonable explanation and supporting references.

AU: thank you for your input. We have improved the discussion part also according to the suggestion of another reviewer. We have deleted the information for the supplier as this was not considered later to focus more on the important parts of the study. The LAB numbers are not evenly distributed in the cooked ham, as the LABs contained in the brine may be injected into the ham before heat treatment and recover during storage sublethally damaged, therefore the LAB numbers are not evenly and regularly increasing as in an artificial contamination experiment.

Another explanation: The absence of microbial growth observed during the storage of ham may be explained by the limited selectivity of the ISO method, particularly with regard to psychotropic bacteria, which are typically underestimated.

#5. Please explain the meaning of NA in Table 4. 

AU: Thank you was explained in Table 4.

#6. In Table 4, Lot 9 had the highest microbial counts in the microorganism in environment, but hams of Lot 9 did not have the highest microbial counts in the storage period among these groups. Please provide possible explanations and supporting references.

AU: Consistent with our findings, Zagdoun et al. [9] also reported fluctuations in LAB counts during the storage period, along with variability the initial concentration at the beginning of the storage due to sublethal damage and long lag phase.

#7. The bacteria count in the Pre-cooking group is higher than the Post-cooking group, which can be considered as a significant reduction of microorganisms in the sterilized ham. However, it is not clear from the experimental methods whether sampling is done in ham production when monitoring microbial levels in the environment. Because it is mentioned P3 Line112-113 that “Swab samples were collected from FPE along the processing line during the production of lot 7 to 9.” This description appears to sampling during the production process. The microbial counts detected in this way should come from the microorganisms contained in the ham itself. But it is mentioned P8 Line75 that “L. carnosum and L. sakei were isolated from the tumbler after cleaning with a combined alkaline detergent-sanitizer (DS)”. This description appears to be a post-production cleaning followed by sampling to verify the number of microorganisms in the environment that may be causing contamination. Please explain in detail the time point of sampling, and explain for these two parts of manuscript.

AU: we included environmental samples during processing (sponges from surfaces) and product-associated samples (brine, surface from tumbled ham) to determine the LAB transfer rate. The transfer rates of LABs from ham in the post-cooking are at the detection limit. The sample from the tumbler after cleaning was intended to determine the presence of LABs, as it is only cleaned once a week.

#8. The bacteria count in the raw meat of Lot6 in Table 2 was not the lowest among all groups. After sterilization, it can be seen in Table 3 that the bacteria count in lot 6 was not the lowest at first, but with increasing storage time, the microbial count in lot 6 decreased and remained very low. Please provide a reasonable explanation and supporting references.

AU: Raw meat introduced into the processing of lot 6 contained no LABs. In the cooked ham samples LAB were also at the limit of detection. Therefore, this lot was of high product quality and did not exceed the limit of 7.4 log CFU/g LAB. If residual flora was present, it was not able to multiply under vacuum packaging.

#9. In P3 Line 65-67, there is mention of the many different spoilage phenomena caused by ham during storage. “(i.e. greenish coloring, discoloration, off-odor, slime production, milky exudates, and gas production)”. In P9 Line 122-123, it mentioned that “in combination with routine sensory evaluation, as these are crucial for determining the end of shelf-life”. However, sensory evaluation is generally used to assess changes in the strength of the sensory characteristics of the product during storage, rather than to assess the undesirable flavors that occur when the ham becomes rancid. It is suggested to revise these sections.

AU: thank you for your considerations. Indeed, sensory testing should be performed during storage of cooked ham and not to determine the deviations at the end of storage. We have changed the discussion and deleted this sentence.

Reviewer 3 Report

The manuscript foods-2427638 entitled " Characterization of Leuconostoc carnosum and Latilactobacillus sakei during cooked pork ham processing”

Introduction:

Introduction: The introduction section does not explain what was already done on topic analyzed. Authors must improve this section by including the relevant information about what has already been achieved on this topic.

Materials & Methods:

What was the criteria to select four different slaughterhouses for Raw meat supply?

Why not a single slaughterhouse?

As per design: 5 lots were provided by supplier A, two lots by B, two by D, but only one lot by supplier C. Why?

Line 80-81: Why the raw meat was stored in the slaughterhouse for 5 days in the box?

Line 85: Why the exact concentrations of brine are undisclosed? I disagree.

This data is being presented for a scientific publication. The concertation of brine is very important for future researchers for the replication of this study.

Please provide the exact concentration of the brine if there is no legal restriction.

Line 92: Please provide specifications of the vacuum packaging equipment and material used for packaging.

Line 92: Where the samples were kept to maintain 4±2 C. Please provide specifications.

How the authors decided that the shelf life of vacuum packaged meat is 40 days even it was processed after 5-6 days of slaughtering. Please provide relevant reference.

Why statistical analysis of the bacterial counts has not been performed/conducted?

Please provide information about statistical analysis.

Results and discussion:

Why the tables and graphs do not have the information regarding level of significance (p- value)?

Please include p- value and superscripts to differentiate the level of significance among different treatments.  

I have read carefully the results and discussion section and I found that the appropriate discussion of the results is lacking. This section needs to be improved further by providing more reference in agreement and disagreement with the results of your study. Also, this section is very disorganized, without any connection between paragraphs with irrelevant comments. Please improve it.

Conclusion: This section is not adequate rewrite please

Author Response

AUTHORS (AU): Thank you very much for your thorough review of the manuscript. We have now revised the manuscript, according to your kind suggestions. The changed sections have been highlighted in yellow.

Introduction:

Introduction: The introduction section does not explain what was already done on topic analyzed. Authors must improve this section by including the relevant information about what has already been achieved on this topic.

AU: thank you for your very relevant comment. We have tried our best to bring the introduction up to date.

Materials & Methods:

What was the criteria to select four different slaughterhouses for Raw meat supply?

Why not a single slaughterhouse? As per design: 5 lots were provided by supplier A, two lots by B, two by D, but only one lot by supplier C. Why?

AU: Answer to all three questions. This study was developed in close collaboration with a meat processing company. The slaughterhouses were not selected for the study, but are in fact the main suppliers to the processing plant. Raw materials are offered by many small and medium-sized suppliers in Austria. In reality, there is not much choice when it comes to raw materials. As we do not refer to or interpret the influence of slaughterhouses on raw material quality in this study, this information has been removed from the manuscript in order to provide more clarity in relation to the LABs.

Line 80-81: Why the raw meat was stored in the slaughterhouse for 5 days in the box?

AU: Thank you for your comment about long storage prior to processing. The raw materials are received and stored at 2-4°C (2-4°C was amended in the text). This is common practice in the meat processing industry in Austria. This is well below the 7°C limit for meat delivery and storage set by EU Regulation 853/2004.

Line 85: Why the exact concentrations of brine are undisclosed? I disagree.

AU: thank you for your input. We have provided the information in materials and methods.

This data is being presented for a scientific publication. The concertation of brine is very important for future researchers for the replication of this study.

AU: thank you for your input. We have provided the information in materials and methods.

Please provide the exact concentration of the brine if there is no legal restriction.

AU: thank you for your input. We have provided the information in materials and methods.

Line 92: Please provide specifications of the vacuum packaging equipment and material used for packaging.

AU: thank you for your input. We have provided the information in materials and methods.

Line 92: Where the samples were kept to maintain 4±2 C. Please provide specifications.

AU: The cooked ham was stored at 4 °C in a temperature-controlled cold room (was amended according to your recommendation).

How the authors decided that the shelf life of vacuum packaged meat is 40 days even it was processed after 5-6 days of slaughtering. Please provide relevant reference.

AU: This study is the result of close collaboration with a meat processing company. the storage of raw material at 2-4°C prior to processing and the determination of a maximum shelf life of 40 days are based on the company's experience and the requirements determined by the retail trade. We have shown in our study that the limit value of 7.4 log is already exceeded after 15-20 days of storage.

Why statistical analysis of the bacterial counts has not been performed/conducted? Please provide information about statistical analysis.

AU: thank you for your recommendation we have applied a binary logistic regression analysis was applied to estimate the probability of reaching the limit of 7.4 log10 CFU/g (AMC and LAB) using storage day as the predictor. A p-value below 5% (p < 0.05) was seen as significant.

Results and discussion:

Why the tables and graphs do not have the information regarding level of significance (p- value)? Please include p- value and superscripts to differentiate the level of significance among different treatments.  

AU: thank you for your recommendation we have applied a binary logistic regression analysis was applied to estimate the probability of reaching the limit of 7.4 log10 CFU/g (AMC and LAB) using storage day as the predictor. A p-value below 5% (p < 0.05) was seen as significant. We have included the information on p-values in the results part. The data set is limited for other statistical analyses.

I have read carefully the results and discussion section and I found that the appropriate discussion of the results is lacking. This section needs to be improved further by providing more reference in agreement and disagreement with the results of your study. Also, this section is very disorganized, without any connection between paragraphs with irrelevant comments. Please improve it.

AU: Thank you for your valuable input. We have tried our best to improve the discussion part according to your recommendation.

Conclusion: This section is not adequate rewrite please

AU: Thank you for your valuable input. We have tried our best to improve the conclusion according to your recommendation.

Reviewer 4 Report

Generally well written research paper on spoilage in cooked, vacuum packed ham. Minor details:

Line 61 define acronym AMC

Line 315 I believe you mean lots not lost

Very minor editing of English needed

Author Response

AUTHORS (AU): Thank you very much for your thorough review of the manuscript. We have now revised the manuscript, according to your kind suggestions. The changed sections have been highlighted in yellow.

Generally well written research paper on spoilage in cooked, vacuum packed ham. Minor details:

Line 61 define acronym AMC

AU: thank you was defined.

Line 315 I believe you mean lots not lost.

AU: thank you was corrected according to your recommendation

Very minor editing of English needed

AU: Thank you very much, we have checked the English spelling and the wording.

Round 2

Reviewer 3 Report

I am satisfied with the authors response.